# Bovine Genital Leptospirosis: An Update of This Important Reproductive Disease

**DOI:** 10.3390/ani14020322

**Published:** 2024-01-20

**Authors:** Luiza Aymée, Julia Mendes, Walter Lilenbaum

**Affiliations:** Laboratory of Veterinary Bacteriology, Biomedical Institute, Federal Fluminense University, Alameda Barros Terra Street, 57, Niterói 24020-150, Brazilju_mendes@id.uff.br (J.M.)

**Keywords:** *Leptospira*, reproduction, abortion, diagnosis, control, streptomycin, vaccination

## Abstract

**Simple Summary:**

Bovine genital leptospirosis (BGL) is an important reproductive disease that leads to embryonic death, gestational losses, stillbirths, or the birth of weak calves. The syndrome is characterized by uterine infection by leptospires, especially from the Sejroe serogroup, which is adapted to cattle. The infection in bulls is still poorly reported. The recommended diagnosis of BGL is the serologic screening of the herds, followed by an individual molecular analysis of genital samples. The control of BGL consists of applying three pillars: vaccination of the herd, treatment of the positive animals, and environmental management. In addition, the application of other procedures such as quarantine of newly purchased animals, and usage of antibiotics in semen diluents or embryo culture media also play a role in the control of the syndrome.

**Abstract:**

Bovine leptospirosis is an important disease that affects the reproductive sphere. Due to its high relevance for the bovine production chain in a worldwide scenario, a better understanding of the disease is crucial to reduce its negative impacts. The main agents are strains from the Sejroe serogroup, such as Hardjo and Guaricura, which lead to renal and genital infection. The genital colonization causes a chronic, silent, and subclinical reproductive syndrome, called Bovine Genital Leptospirosis (BGL). Embryonic death, estrus repetition, subfertility, and abortions are the main signs of BGL condition in females. However, although leptospires have been identified in semen, the manifestation of BGL in bulls remains to be clarified. The recommended diagnosis of BGL includes a serologic screening of the herds using the microscopic agglutination test followed by PCR of genital samples (cervicovaginal mucus, uterine fragment, or semen), especially from animals with reproductive failures. After the identification of carriers, control is carried out considering three steps: antimicrobial treatment of the carriers, environmental and reproductive management, and herd vaccination. Systematic testing, quarantine of newly arrived animals, and usage of antimicrobials in semen diluents or embryo culture media are other sanitary approaches that are encouraged to improve the control of the syndrome. Herein we discuss protocols for an efficient diagnosis and preventive procedures of BGL, which are fundamental to reducing the negative impact of the disease on cattle reproduction and its consequent economic hazards.

## 1. Introduction

Reproductive diseases are one of the main causes of hazards and lack of productivity in dairy and beef cattle production. It is estimated that up to 50% of embryonic deaths in bovines are associated with infectious diseases [1]. In that context, leptospirosis is one of the major causes of reproductive failures in ruminants, which leads to abortions, estrus repetition, stillbirths, the birth of weak calves, a decrease in growth parameters, and a drop in milk production [2,3]. The striking economic impact of bovine leptospirosis was estimated at USD 97 to USD 2611 per abortion [4]. In addition, an outbreak can lead to an annual hazard of up to USD 150,000, including abortions and costs with preventive measures, as occurred in Argentina [5]. Although those studies calculated the costs of abortions, the hazard caused by the subclinical manifestation of leptospirosis remains to be calculated. In that context, an earlier study evaluated that overall poor reproductive performance in cows may cost approximately €231 for each cow per year [6]. Another study demonstrated that the hazard of cows with pregnancy losses can reach up to USD 2000 per loss, considering also the extended non-pregnant period and the risk of early culling [7]. Since subclinical disease is the most common manifestation of leptospirosis in cattle, the economic impact of these features in a herd is probably impressively higher than simply the cost of abortions. Those negative outcomes of leptospirosis can be reduced by the correct diagnosis of the disease, followed by adequate preventive measures. Recent studies advanced the knowledge of leptospiral colonization in cattle, as well as the main etiological agents and reproductive manifestations, particularly of naturally infected animals in field conditions. Based on this, the purpose of the present study was to gather and critically analyze the current information regarding the role of bovine genital leptospirosis as a reproductive disease.

## 2. Leptospirosis in Cattle

### 2.1. Agents and Epidemiological Aspects

Bovine leptospirosis is a worldwide disease caused by several strains of the spirochete bacteria *Leptospira* spp. [8]. Leptospires can be classified by genetic or phenotypic features [9]. In its genetic classification, the *Leptospira* genus is divided into approximately 68 species, which are classified as pathogenic (subclade P1), intermediate (subclade P2), and saprophytic (subclades S1 and S2) [10]. Regarding the phenotypic classification, also known as serological, the leptospiral strains are divided based on the characteristics of the lipopolysaccharides (LPS) of their outer membrane [11]. In this serological division, the strains are classified into approximately 300 serovars, which are grouped into 24 serogroups according to the homogeny of their LPS [12].

The main known agents of bovine leptospirosis are the strains of Sejroe serogroup, which are considered adapted to the bovine species and do not require other reservoirs for their maintenance, being transmitted from cow to cow [9,13]. Sejroe strains have been also isolated worldwide, besides being frequently demonstrated in serosurveys, and in molecular studies [14,15,16]. Additionally, this serogroup has been associated with reproductive failures [3,17], which means that Sejroe strains play a key role in the reproductive syndrome of bovine genital leptospirosis (BGL). Despite its adaptation to bovine, the importance of these strains can be attributed to the preference of Sejroe for colonizing the uterus [18]. The main serovars of this group are Hardjoprajitno (*L. interrogans*) and Hardjobovis (*L. borgpetersenii*) [14]. Additionally, serovar Guaricura (*L. santarosai*), also belonging to the Sejroe serogroup, has been frequently identified in South American cattle [17,19]. However, the dynamics of genital infection, adaptability to the host and the real effects of infection in cattle by serovar Guaricura are still not completely understood. Other agents of bovine leptospirosis are the incidental strains Icterohaemorrhagiae, Pomona, Australis, and Grippotyphosa (*L. interrogans*), which are adapted to other hosts but still can infect cattle [8,20]. Although incidental infection in bovine is less frequent than Sejroe infection, it can lead to acute manifestation, especially in calves, and abortion outbreaks [9].

The transmission of leptospires, regardless of its host origin, usually occurs by shedding bacteria in the urine and other fluids of the infected animal [15,20], and the susceptible bovine will be infected by its contact with the contaminated environment. Sejroe strains are transmitted from cattle to cattle, probably not only by a contaminated environment but also during mating. Sexual transmission in cattle has been widely suggested, occurring in male to female direction and vice versa [13,21,22] since the presence of pathogenic leptospires has been reported both in male and female reproductive tract [17,23,24].

### 2.2. Pathogenesis of Leptospiral Infection in Bovine

The clinical aspects of bovine leptospirosis indicate that it is an infectious disease involving the reproductive system. Paradoxically, most of the research carried out in cattle focuses on the detection of renal carriers, while genital carriers of leptospires are neglected. For a long time, infection in the reproductive tract was considered to be secondary to renal pathogenesis in chronic infection, and the kidneys were considered the main immunologically privileged site [25]. However, the presence of leptospires in ovaries, ovarian follicles, and oviducts has already been demonstrated [26,27], as well as in the uterus [17], cervicovaginal mucus [24], and semen [23], which confirmed the genital tract as an important extra-renal location of leptospires. In addition, a relevant immune response in the reproductive tract concomitant with low serum antibody titers has already been recognized [28,29]. Additionally, experimental infection demonstrated long-term colonization of the genital tract by Sejroe strains [30]. Although little is known about the pathogenesis of leptospiral strains in those reproductive sites, it is suggested that reproductive failures would be a direct effect of this genital infection and long-term colonization. Moreover, sexual transmission both in the male-female direction and in the opposite direction has been suggested [9,13,23,31], and uterine infection has been experimentally demonstrated [30]. Still in this context, recent studies show that leptospires of the Sejroe serogroup, adapted to cattle, preferentially colonize the reproductive tract, which seems to be their main colonization site [18,32]. In a recent review, this evidence was compiled, and the “Bovine Genital Leptospirosis” (BGL) syndrome was suggested as a disease dissociated from renal infection, rather than a secondary infection [13]. After that, the study of genital infection (even in other animal species) was performed by ours and other groups to confirm its agents, understand pathogenesis, and improve diagnosis methods and control protocols [33,34,35,36]. 

#### 2.2.1. Pathogenesis and Clinical Manifestation in Females

The main signs of BGL are pregnancy losses, which range from embryonic death (more frequently) to late-term abortions (Figure 1). The embryonic mortality is classified as early embryonic death (EED) when it occurs until 28 days after fertilization, while the late embryonic death (LED) includes the period of 29 to 45 days [37]. In addition, there are pivotal periods in the initial pregnancy trimester [38] that can lead to the death of the embryo/conceptus: (a) Pivotal Period 1, leading to failure in fertilization in the first week after breeding; (b) Pivotal Period 2, which occurs between day 8 to 27, leading to error in the maternal recognition of pregnancy; (c) Pivotal Period 3, which is characterized by the placentome development between days 28 and 60 after breeding. 

Regardless of the diagnostic method, it is quite difficult to identify the embryonic death in the field, as well as the exact moment that it happened, especially in the pivotal periods 1 and 2. The cows that suffered EED until day 16 after breeding will return to their cycle regularly, while the cows that suffered losses after this period might be related to irregular return to estrus [37,38]. Due to the challenging diagnosis of those pregnancy losses, the precise pathogenesis of how leptospires lead to embryonic death is still unclear. Loureiro and Lilenbaum [13] hypothesized two mechanisms, that can occur either separately or combined: (a) the direct infection of the embryo, leading to its non-viability. In this case, embryo death could occur in Pivotal Period 1 because of the direct damage caused by leptospiral infection. The embryo infection by leptospires had been already demonstrated by the penetration of zona pellucida in an in vitro study [24] and was later reinforced by the presence of leptospiral DNA in the follicular fluid of naturally infected cows [27]; or (b) the inflammation of the endometrium caused by leptospiral infection [39] impairs the uterine environment and the embryonic development, being related to the Pivotal Period 2. 

Although the presence of leptospires in the uterus of naturally infected cows has been reported [17,22], the exact effect of the infection on embryo development remains to be elucidated. In the physiology of pregnancy, the integrity of the endometrium and its function are pivotal to the nourishment and attachment of the embryo [40]. In that context, the oviducts and endometrium secrete a complex fluid, termed histotrophs, that nourishes the conceptus, being its’ only source of nutrients until implantation, usually on the 20th day after fertilization in cows [41]. Previous studies demonstrated that disturbances of the histotroph composition or production can impair pregnancy establishment [40], and other bacterial infections, such as *Escherichia coli* or *Trueperella pyogenes* in the endometrium can alter the composition of these important secretions [42]. In that context, although the uterine immune response to the infection by *Leptospira* spp. is still unclear, we hypothesize that one of its effects could be the dysregulation of the histotroph composition or even the reduction or compromising of its secretion. That hypothesis supports the role of the alteration of the uterine environment leading to embryonic death, especially in early gestation. 

Since BGL is a chronic infection, subclinical signs can remain for years causing long-term subfertility, and therefore, leading to extensive economic hazards in the herd. In that context, the cow will be classified as a repeat breeder due to frequent embryonic death, presenting subfertility, or even infertility. A direct effect of embryonic mortality is the reduction in the conception rates, which certainly will impair the profitability of the herd [37,43]. Other signs that are a consequence of leptospiral infection but occur less frequently than the subclinical signs are abortions, stillbirths, or the birth of weak calves. Abortions have been widely associated with bovine leptospirosis, perhaps the most visible sign, being linked to incidental strains when occur as an outbreak, or to adapted strains when occur endemically [44]. It was suggested that abortions by leptospiral infection occur in the final third of pregnancy; however, fetal death has been reported in all stages [45]. The pathogeny of fetal death is also unclear, but the direct infection of the fetuses is the most probable cause. The lesions found in aborted fetuses positive to leptospires are widely variable, affecting different organs at different levels. The abortions infected by incidental strains such as Pomona or Icterohaemorrhagiae usually present icteric and leptospires may be demonstrated in the kidneys and liver [46]. In contrast, anicteric fetuses with hemorrhagic or even presenting no visible lesions are often associated with infection by Sejroe, which may be demonstrated in several organs, such as cardiopulmonary tissue, thymus, subcapsular kidney content, and abomasal liquid [45].

Besides abortions, stillbirths and the birth of weak and lighter calves also occur as a consequence of bovine leptospirosis [9]. In the case of leptospirosis in horses, the birth of weak foals is associated with placentitis [47]. The inflammation of the bovine placenta has not yet been associated with genital infection by leptospires, and further studies should be performed to understand if placentitis could play a role in bovine abortion pathogenesis. The birth of a dead fetus before or during calving at full term is defined as stillbirth [48]. The exact mechanism of how it happens has not been fully understood. All those clinical signs have the same pathogenesis as the abortions, being caused by the fetal intrauterine infection, but in this case, in late gestation [9].

#### 2.2.2. Aspects of Genital Infection in Males

Males can also carry leptospires in their reproductive tract. Genital infection has been identified in bulls [21,23], semen of rams [33] bucks [49], boars [50], and camelids [51]. Nevertheless, the effect of genital infection by leptospires in males is still poorly understood, and consequently, leptospiral infection in bulls has been highly neglected. In a recent systematic review about the role of bulls’ infectious infertility, leptospirosis has not even been mentioned [52], reflecting its underestimation. Conversely, in a recent critical review, leptospirosis was classified as a potentially transmitted disease of bulls [53], albeit it was less emphasized than other infectious diseases such as trichomoniasis and campylobacteriosis. Although the presence of leptospires in semen seems clear, the real effects of this infection on the semen quality and fertility of the bulls have not yet been clarified. One single study enlightened this issue, showing a correlation between the infection of naturally infected bulls seroreactive against Sejroe with the presence of necrospermia or azoospermia [54]. However, although there is some evidence that leptospiral infection can impair semen quality, more research should be performed in this field. If the infection indeed jeopardizes the semen quality, a new concern is added to the presence of BGL in bulls, besides their crucial role in the epidemiology of the syndrome.

## 3. Diagnosis

The recommended methods to diagnose leptospiral infection in cattle are the indirect serological method microscopic agglutination test (MAT), and direct molecular techniques such as conventional (PCR) or quantitative polymerase chain reaction (qPCR). While MAT only predicts a previous exposure of the animal to the pathogen, direct methods such as culturing or PCR indicate the presence of infection [55]. 

### 3.1. Microscopic Agglutination Test (MAT)

MAT is indicated by the World Organisation of Animal Health and the World Health Organization for the diagnosis of animal and human leptospirosis and is particularly useful in diagnosing acute diseases or outbreaks [56,57]. The test consists of measuring the agglutinating antibody titers against a panel of live leptospires, representing different serogroups [58]. Based on this, it is possible to perform a presumptive diagnosis of the serogroup, but not the serovar, involved in the infection [50], and estimate the level of infection in the herd. Nonetheless, this serologic test presents limitations for the diagnosis of bovine leptospirosis. In cattle, since the acute presentation of leptospirosis is rare, the MAT may not be adequate and is only reliable at a herd level, not being recommended for individual diagnosis [59,60]. As the infection in bovine is chronic and determined by adapted strains, it is frequently related to low titers of systemic antibodies; thus, many chronically infected animals may not present seroreactivity [14]. Furthermore, it is not possible to distinguish vaccinal antibodies from those that originated from an infection, and this is a method whose analysis is subjective [58]. Due to the limitations of this method, MAT is recommended as a first step in the diagnosis, for the herd screening (at least 90 days after the last vaccination) to assess the status of leptospirosis in those herds. In our laboratory, since we are in an endemic tropical area, we require >10% of the tested animals seroreactive with titers ≥200 to consider the herd as seroreactive. In this case, a second confirmatory step by PCR of the animals with reproductive failures is performed to detect genital carriers at a cow level and initiate their treatment [14].

### 3.2. Molecular Diagnosis

Although culture is commonly used to diagnose bacterial diseases and is considered the gold standard for leptospirosis diagnosis [9,14], it is not recommended for diagnostic purposes. Leptospires are fastidious and very nutrient-demanding bacteria; therefore, their culture takes too long and presents low sensitivity [61]. On the other hand, molecular techniques such as PCR and qPCR present high sensitivity and specificity [62], are fast to conduct [63], and can be performed in various types of clinical samples such as urine, cervical-genital mucus, endometrial biopsy, semen, fetal tissues, and placenta [17,22,45]. It is noteworthy that the second step for diagnosis, i.e., the molecular investigation in genital samples, is indispensable for a reliable diagnosis since chronic reproductive disease due to genital leptospiral colonization is very often not detected by serology [13].

The association of herd serology with *lip*L32-PCR of genital samples, particularly cervicovaginal mucus (CVM) of the cows, has been an efficient protocol for identifying leptospiral carriers [17]. The CVM is a mirror of the uterine environment, and its sampling is a simple and inexpensive technique that is performed with the use of a vaginal speculum and a cytologic device with a cytology brush in the extremity. The brush is rubbed in the vaginal fornix to collect the mucus originating from the uterus, avoiding urine contamination [13]. The detection of genital carriers by molecular analysis of CVM was demonstrated to be more suitable than the usage of urine [17]. In that context, the identification of the carriers is crucial to implementing an accurate preventive approach and reducing the impact of leptospirosis on the reproductive parameters of the herds and their profitability as well.

## 4. Control Measures

The control of leptospirosis is commonly based on three aspects: antimicrobial treatment, environmental control, and vaccination (Figure 2), preferentially implemented altogether [64]. 

### 4.1. Treatment

The treatment with antimicrobials is performed to eliminate the carrier status of the animals diagnosed with leptospiral infection. It is recommended that the treatment of positive animals must be performed at the early beginning of the control program, to reduce the shedding of leptospirosis in the body fluids and avoid new infections in animals of the same herd [9,65]. The administration of antimicrobials is also recommended as one of the quarantine procedures for newly arrived animals in the herd [66]. The use of streptomycin in the dosage of 25 mg/kg has been for a long time the recommended protocol to treat leptospiral carriers [67]. However, the number of administrations needed can vary according to infected sites: while renal infections can be successfully treated with a single dose [67], it has been demonstrated that three doses of streptomycin may be required for genital treatment [68]. The treatment of positive bulls, as well as the addition of antimicrobials in the sperm extenders of frozen semen, is also important to control the suggested sexual infection [21]. However, although streptomycin protocols have demonstrated efficiency in treating carriers, the prohibition of the use of this drug in the USA and Australia is a critical point [69]. Studies involving other antimicrobial drugs, such as enrofloxacin [65], oxytetracycline, and ceftiofur [69] have demonstrated promising results regarding the treatment of renal carriers but have never been tested for genital colonization. The treatment of genital colonization with streptomycin requires a great quantity of the drug, for three consecutive days [68], besides the withdrawal period with milk discard, increasing the costs [64]. In that context, we strongly encourage the search for other antimicrobials and alternative protocols for the elimination of leptospires in the reproductive tract of bovines.

### 4.2. Environmental Control

Another essential step in controlling leptospirosis is environmental control. Due to the survival of the bacteria in the environment and the high diversity of animal carriers, both domestic and wild animals, environmental control is particularly challenging, especially in tropical regions. Although it is widely known that rainy seasons facilitate the occurrence of outbreaks caused by incidental strains [66], it does not appear to be striking in infections by adapted strains such as BGL, which can occur independently of the environmental conditions [70]. Additionally, incidental infection might also be promoted by shared grazing with different animal species, such as swine (reservoir of the strain Pomona) or wildlife, but it does not seem to be a determinant for BGL transmission. 

### 4.3. Vaccination

In any case, the results of a control program can be boosted by the immunization of the herd [71]. The vaccination represents the third aspect of the triad, indeed the cheapest and most employed measure, and shall be conducted periodically in the whole herd [64]. Additionally, the vaccination of the animals before their introduction in endemic areas is highly recommended [15]. Currently, despite a lot of research regarding recombinant vaccines, the available commercial vaccines are all composed of bacterins, i.e., the whole inactivated bacterial cell [72,73]. The bacterins lead to an increase in humoral and cellular immune responses, being particularly directed against the serogroups present in the vaccine composition [74,75]. Although some discussion about serovar specificity has been raised, several studies showed satisfactory homologous protection between serovars of the same serogroup and, less frequently, between different serogroups [69]. Certainly, representatives of the main serogroups that occur in the region must be in the vaccine’s formulation, and epidemiological vigilance by serology plays a strategic role in this part. Importantly, commercial vaccines are currently formulated with strains of international reference, which might not be the best choice to prevent infection by local strains; thus, research with bacterins containing local or at least regional strains must be strongly encouraged [64]. Although it slightly varies with the recommendations of the manufacturer, the immunization of bovines generally occurs in a single dose annually [76], and it is recommended to be performed immediately before the reproductive season [1]. Depending on the endemicity of the herds’ region, two or even three doses per year have been employed. Regarding the effect of vaccination on the reproductive parameters, studies have shown that vaccinated herds present have fewer repeat breeders [3], and higher pregnancy rates after artificial insemination [1,77], reducing economic hazards. Despite the improvement in those parameters, studies showed that vaccination cannot prevent colonization, neither in kidneys [76] nor the genital tract [30], and infected animals can keep shedding leptospires, which reinforces the need for permanent sanitary control. 

### 4.4. Other Sanitary Measures

Since the transmission of BGL is little affected by environmental conditions, specific sanitary actions mainly focused on reproduction take place as the major control procedures. Molecular screening using genital samples is crucial to diagnose the carriers, which must be isolated and treated. Testing of newly arrived animals in the herd is mandatory, since the introduction of infected animals, especially by Sejroe strains, can compromise the controlled status of a herd [78]. The biosecurity procedures shall also be applied by the testing and treatment of positive bulls, even those without clinical manifestation, especially in recently acquired or leased animals [79]. Natural breeding should be performed only with negative bulls and periodic testing (PCR) should be implemented to prevent sexual transmission. Similarly, the semen used for artificial insemination should receive the addition of antimicrobials in the diluents, avoiding the infection of the inseminated cows [21]. The inclusion of antibiotics such as penicillin, streptomycin, or gentamicin as a supplement in the embryo culture media or embryo wash solution can also remove the contamination [80,81]. Additionally, the usage of antimicrobials is recommended by the guidelines of the International Embryo Technology Society (IETS) and the World Organisation of Animal Health (WOAH) [82,83].

## 5. Conclusions

Bovine Genital leptospirosis is an important chronic neglected disease, which leads to subfertility and remarkable economic and sanitary hazards to the reproduction of cattle, despite its silent and inapparent form. Knowledge about its etiology and the adequate usage of new tools for the proper diagnosis, mainly molecular detection of genital carriers, is mandatory to efficiently implement the control procedures. Although several gaps remain in the understanding of BGL, many advances toward the diagnosis and control of BGL have been made in the last few years. In that context, improving diagnosis tools and developing appropriate control procedures specifically directed towards BGL is fundamental to reducing the negative impact of the disease on cattle reproduction.

## Figures and Tables

**Figure 1 animals-14-00322-f001:**
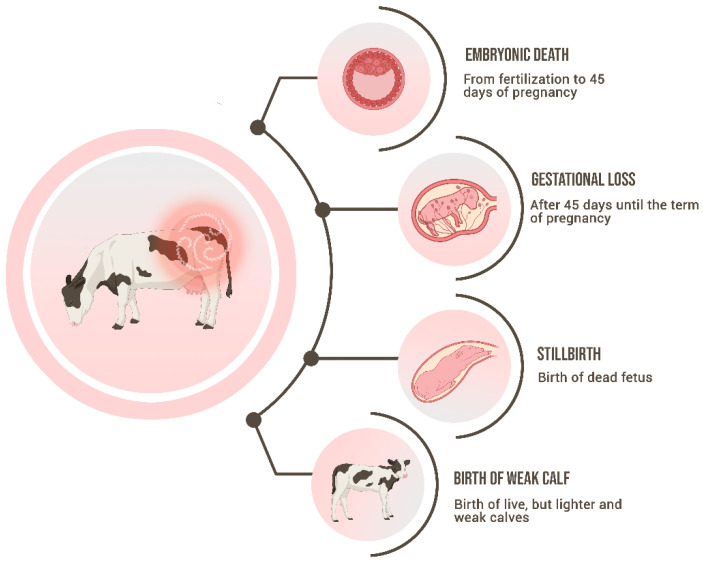
Schematic representation of the main signs of Bovine Genital Leptospirosis.

**Figure 2 animals-14-00322-f002:**
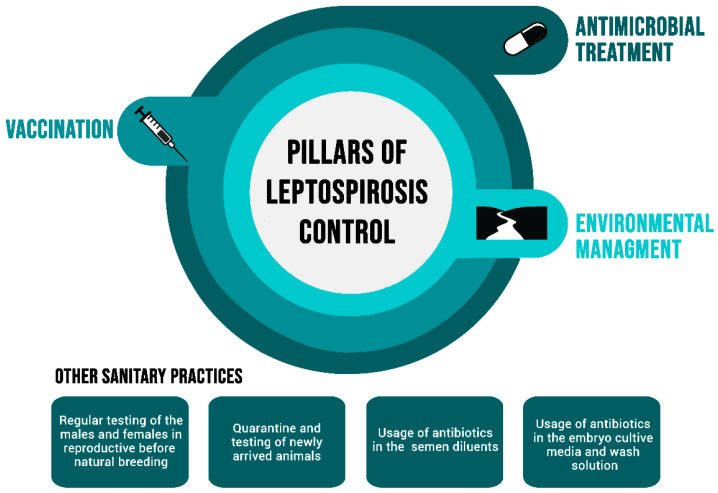
Schematic representation of the three pillars of control of bovine leptospirosis and the other sanitary practices recommended to prevent BGL.

## Data Availability

Data are contained within the article.

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
