# Peer review of "Bovine Genital Leptospirosis: An Update of This Important Reproductive Disease"

_animals, 2024, doi:10.3390/ani14020322_

Round 1

Reviewer 1 Report

Comments and Suggestions for Authors

Brief summary:

The present paper is aimed at providing an update on bovine leptospirosis, particularly regarding the genital form of the disease. Bovine leptospirosis is a neglected disease in cattle farming and not yet thoroughly studied. Moreover, the spread of leptospirosis, in both animals and humans, could be also significantly affected by global changes which act on the environment. Therefore, this disease deserves full attention and it is important that new scientific contributions are produced.

 Review:

The paper describes leptospirosis in cattle, analyzing in particular the pathogenetic and clinical aspects of the genital form (previously referred to as Bovine Genital Leptospirosis: see reference n. 10), in addition to the diagnostic and control aspects. This review follows a previous recent review on Bovine Genital Leptospirosis (BGL), in which one of the authors was involved (see reference n.10, year 2020). The present review presents several parts similar to the previous review (and partly also to the review reference n. 55, year 2017, with the same author), particularly for the sections concerning the diagnosis and control of the disease. Figure 2 is also partially superimposable on figure 1 of reference n. 55. However, approximately 45% of the bibliography referred to in the text (of which approximately 16% was produced by the research group of the authors) was published after reference n. 10. Therefore, the paper constitutes, as rightly indicated in the title, an update on the BGL, supported by an adequately updated bibliography. The number of self-citations is significant, but justified by the scientific production of the research group to which the authors belong. The manuscript is well-written and easy to understand.

 Specific comments:

Page 2, lane 47: “US$ 150,000” globally or for each farm or what? Please clarify/specify.

Page 2, lane 52: “US$ 2,000” per cow? Please clarify/specify.

Page 2, lanes 73-74: from reference 11 directly to 13, missing reference n. 12.

Page 2, lanes 90-93: “For a long time, infection in the reproductive tract was considered to be a secondary effect of renal pathogenesis, occurring as a result of occasional septicemia after a previous renal infection [21]”. The sentence is not supported by the reported reference, which does not refer to the pathogenesis of the genital form (which is not yet sufficiently clarified), but only discusses host-pathogen interactions and the pathogenesis of the disease in the kidney. In my opinion, in the present form the sentence (especially the second part of the sentence) suggests that it was believed that the infection of the reproductive tract follow the renal infection, but this belief seems unlikely for at least two already known considerations:

1) that the infection is followed by an initial bacteremic phase and the dissemination to the renal tissues is haematogenous. As also reported by the reference indicated, it has been observed that in the rat L. Copenhageni is disseminated extensively to tissues (to all examined tissues: kidney, spleen, lung and liver) during the early stages of infection. This is followed by a clearance of Leptospira from all tissues (likely due to the rise of anti-leptospiral immunoglobulins), except kidney. We cannot exclude that the same dynamic also occurs for the reproductive tract;

2) that the persistent presence of a certain degree of immunity (although variable) following the infection, suggest that a septicemic event following the renal infection could only be occasional, as rightly pointed out by the authors. But this seems discrepant with the frequency of genital infection found in the bovine species.

Please rephrase or reconsider the sentence.

Page 3, lanes 103-104: “After that, many groups began to study BGL, including its agents, pathogenesis, diagnosis methods, and control protocols”, please add references to this sentence.

Reviewer 2 Report

Comments and Suggestions for Authors

“Bovine genital leptospirosis: an update of this impacting reproductive disease”

The authors have described the cause, manifestation, diagnosis and control of bovine reproductive leptospirosis. The article is fairy well-written, but there is still need for improvement in terms of organisation and content of the article. Moreover, the authors should cross-check the manuscript in a thorough manner, and revise accordingly the various grammatical errors.

Introduction: The real motivation or justification for writing this review article should be highlighted. In lines 47 and 48, the authors have stated that the impact caused by subclinical manifestation of Leptospira infections has not been calculated yet; does this mean that the reason for writing the article was to calculate the impact of subclinical infections? By reading through the article, this does not seem to be so.

The title reads “an update of this impacting reproductive disease”. I would have expected a section that clearly highlights the documented occurrences and impact of the disease.

Section 2 “Leptospirosis in cattle”. This section is a mix of different content. What is this section about? Reading through, I see content on cause/taxonomy, manifestation, transmission, research gaps, pathogenesis, immune response. All this in one section. The authors should re-organize the article to have these topics described in a systematic manner.  I would expect a section on the epidemiology of the disease.

Section 4: Diagnosis. This section should be sub-divided to assess the different diagnostic methods available for the disease.

Section 5. This should be “Prevention, treatment and control”, and the three aspects should be described separately with relevant sub-headings.

Comments on the Quality of English Language

The language quality can be improved

Round 2

Reviewer 2 Report

Comments and Suggestions for Authors

The manuscript has improved following the author’s revisions.

Here are minor comments to be addressed by the authors

L21: It should “bovine production chain”

L65: Delete the colon in the heading. Do the same to other headings in the document.

L66: Delete comma. Replace “determined” with “caused”

L80: Add comma after the brackets

L93: Replace “the shed” with “shedding”

L94: Replace “by” with “in”

Comments on the Quality of English Language

Minor comments, as presented above

Author Response

Dear reviewer, we have modified the manuscript according your suggestions. Thank you again for your contribution.